# Characteristics of Children with an Undesirable Motor Competence Development During the Transition from Early to Middle Childhood: Results of a 2-Year Longitudinal Study

**DOI:** 10.3390/ijerph21111460

**Published:** 2024-10-31

**Authors:** Pim Koolwijk, Ester de Jonge, Remo Mombarg, Teun Remmers, Dave Van Kann, Ingrid van Aart, Geert Savelsbergh, Sanne de Vries

**Affiliations:** 1Research Group Healthy Lifestyle in a Supporting Environment, The Hague University of Applied Sciences, 2501 EH The Hague, The Netherlands; e.a.l.dejonge@hhs.nl (E.d.J.); s.i.devries@hhs.nl (S.d.V.); 2Institute of Sport Studies, Hanze University of Applied Sciences, 9747 AS Groningen, The Netherlands; r.mombarg@pl.hanze.nl (R.M.); i.van.aart@pl.hanze.nl (I.v.A.); 3School of Sport Studies, Fontys University of Applied Sciences, 5644 HZ Eindhoven, The Netherlands; t.remmers@fontys.nl (T.R.); d.vankann@fontys.nl (D.V.K.); 4Department of Health Promotion, Maastricht University, 6229 HA Maastricht, The Netherlands; 5Department of Human Movement Sciences, Section Motor Learning & Performance, Vrije Universiteit (VU) Amsterdam, 1081 HV Amsterdam, The Netherlands; g.j.p.savelsbergh@vu.nl; 6Department of Public Health and Primary Care, Health Campus The Hague, Leiden University Medical Centre, 2511 DP The Hague, The Netherlands

**Keywords:** early childhood, middle childhood, motor competence development, perceived motor competence, enjoyment, sports participation, body mass index

## Abstract

Objective: Motor competence development from early to middle childhood is accompanied by great variance. This course can be influenced by many factors in the ecosystem. The objective of this study was to examine which individual characteristics are associated with **an undesirable** motor competence development during the transition from early to middle childhood. Methods: A longitudinal study was conducted between February 2020 and May 2022. Actual and perceived motor competence and the potential determinants physical activity enjoyment, weight status, and organized sports participation of children (**49% boys**) aged 4–6 years old at T0 (N = 721) were measured at two points in time, separated by a two-year interval. Associations between potential determinants and AMC, including interactions with time, were analyzed using linear mixed-effect regression models with continuous motor quotient scores as outcome variables. Results: Overweight, obesity, and lack of organized sports participation were associated with lower motor quotient scores over time. Multivariate analyses showed that associations of weight status (overweight and obesity) and sports participation with motor quotient scores remained significant after adjustment for variations in perceived motor competence and physical activity enjoyment. Conclusions: Excessive body weight and lack of sports participation from early childhood are associated with an increased risk of an undesirable motor competence development over time.

## 1. Background

Early childhood (2- to 5-years of age) is characterized by rapid growth and development. It is considered a critical period for establishing healthy behaviors, such as physical activity [1,2,3]. One potential factor underlying participation in sports and physical activity is motor competence [4,5,6,7]. Motor competence (MC) is defined as an individual’s degree of proficiency in performing a wide range of motor skills as well as the mechanisms underlying this performance (e.g., motor control and coordination) [8]. Clarke and Metcalfe [9], in 2002, introduced a well-recognized model, i.e., the mountain of motor development, in which the different stages of MC development are described. This ‘mountain’ includes an ascent of six passages of MC development: the reflexive period, the preadapted period, the fundamental patterns period, the context-specific period, the skillful period, and the compensation period. Each period builds on the skills learned in the previous period. In their model, MC development is considered a nonlinear process, as the learning process within each child does not develop smoothly from one stage to the next and differs in pace. Previous studies have shown that a significant number (20–30%) of children between early childhood (EC) and middle childhood (MCD) (6- to 9-years of age) exhibit undesirable MC development [10,11,12,13,14,15]. Although children may naturally develop rudimentary motor skills through maturation, achieving proficient forms of skills may only be achieved in a developmentally appropriate environment. During EC, physical activity (PA) influences the development of MC through a variety of movement experiences, and vice versa [7,16]. However, increasing the amount of PA does not guarantee the gradual and positive development of MC [3,13,17,18]. MC development is also influenced by other determinants. These determinants can be categorized into individual characteristics (e.g., gender, ethnicity, age, PA), education-related programs (intervention programs for stimulating MC), social environments (e.g., parental style and family-related variables), and physical environments (e.g., traffic infrastructure, playground facilities) [19,20,21]. In this manuscript, we focus on the individual characteristics of MC development.

Regarding the individual characteristics that may influence MC development, an Australian study showed that children in low-income communities, especially boys and underweight and overweight children, have higher odds of being at risk for a delay in MC [22]. Weight status (quantified by body mass index (BMI) [8,23] and perceived motor competence (PMC) [24,25,26,27] are, in general, well-recognized determinants for MC development in either EC or middle childhood (MCD). Several studies have shown a negative relationship between BMI and MC [28,29,30]. Recently, an international motor development consortium analyzed the association between BMI and MC in a large multicountry sample of EC-aged children (*n* = 5545) [31]. A negative association was found between BMI and overall (locomotor and object control skills) MC [31]. Other, longitudinal, studies also found that the weight status of EC-aged children is an important predictor of children’s MC in MCD [23,32]. Evidence regarding this negative association was strengthened by a systematic review performed by Barnett et al. [24] in which longitudinal and experimental evidence in support of Stodden’s conceptual model was evaluated [33]. In this model, it has been hypothesized that the synergistic relationship between MC and PA is mediated by PMC, health-related fitness variables, and weight status, with this relationship expected to strengthen over time [33]. In the review by Trecroci and colleagues [34], the relation between actual motor competence (AMC) and PMC with weight status in MCD children was investigated. A negative association was found between AMC and PMC and weight status. However, after the risk of bias was assessed, the level of evidence linked to this association remained uncertain and lacking.

In the context of this study, PMC is used to refer to an individual’s awareness and belief in his/her capabilities in goal-directed human fundamental movement skills and what would be considered as MC [35]. Children with lower PMC are at risk of losing motivation to participate in movement-related tasks and reduce PA engagement [36]. PMC mediates the relation between AMC and children’s level of PA during EC and MCD [24,27,37,38,39]. However, during the early years, this relationship is not always that strong [36], and sometimes results are mixed [40,41]. Stodden (2008) [33], later revised by Robinson and colleagues (2015) [6], suggested that as children age and cognitively develop, the relationship between their actual MC and their perceived MC becomes stronger. However, longitudinal evidence is needed to confirm this relationship in the transition from EC to MCD.

In addition to children’s perception of their motor competence, enjoyment of physical activities may also be an important individual characteristic of prolonged participation in sports and physical activities [42,43]. Especially in EC, PA drives the development of MC, whereas proficient levels of MC will increase PA during MCD [33]. It has been argued that the execution of motor skills on its own is insufficient for motor learning and its development if it is not linked with positive emotions [44,45]. Studies show that children who enjoy participation in physical education (PE) are more physically active than those who do not enjoy PE [46,47,48,49]. In line with this, one would expect an association between the enjoyment of physical activities with MC development. However, to our knowledge, the relationship between enjoyment of PA and MC development over time is still understudied.

Further, by the time children go to school (at age 4 within the Dutch school system), education-related programs such as physical education (PE) lessons and sports clubs are important for MC stimulation [50,51,52]. A recently conducted Dutch study indicated that determinants in the social and physical home environment and direct living environment are associated with MC disparities during EC. Additionally, parenting practices and parental PA involved behaviors are relevant modifiable factors related to differences in children’s MC [53]. Finally, excessive screen media use has been associated with poor MC development and decreased PA in EC-aged children, especially among those with prolonged exposure [19].

Thus, longitudinal evidence regarding individual characteristics of children during the transition from EC to MCD is needed [54,55]. Besides more longitudinal evidence during this transition phase, establishing risk profiles of children from whom MC development is lagging behind is crucial for developing targeted interventions. To date, undesirable patterns of MC development associated with individual characteristics such as weight status, PMC, PA enjoyment and sports participation have only been examined in mostly cross-sectional studies with relatively small sample sizes. Therefore, the objective of this study is to examine which individual characteristics are associated with an undesirable MC development during the transition from early to middle childhood. Based on cross-sectional studies, it is expected that children with an undesirable MC development are characterized by a higher BMI, lower PA enjoyment scores, and lower organized sports participation. Regarding PMC, lower scores are expected for children who exhibit undesirable MC development.

## 2. Methods

### 2.1. Study Design and Participants

For the purpose of this study, longitudinal data from the ‘Start(V)aardig’ project (Dutch for ‘Skilful Start’) were used. These data were gathered among a convenience sample of children aged 4–6 years old from 36 primary schools across the Netherlands with variations in geographic area (i.e., rural/city), school types (i.e., Catholic/public) and socioeconomic position (i.e., six categories ranging from low ≤ −0.20 to high ≥ 0.20). Numerical indicators of the socioeconomic position of the school region were based on wealth, education level, and employment history of households) [56]. The participating schools were all internship schools of connected universities. AMC, PMC, PA enjoyment, weight status, and organized sports participation of children aged 4–6 years old were measured at two points in time, separated by a two-year interval (i.e., baseline and one follow-up measurement). All assessments were performed during a regular 50-minute PE lesson at school. Baseline data (T0) were collected between February and May 2020, covering a total sample of 1128 children (5.35 years of age (SD ± 0.69), with 50% boys). The first follow-up measurement of AMC was conducted between February and April 2021 (T1), and the second follow-up measurement was conducted among 721 children (7.48 year of age (SD ± 0.62), with 49% boys) between March and May 2022 (T2). These data were used to determine undesirable patterns of MC development over time [14]. Parents gave written informed consent for their children’s participation (63% consent rate). This study received ethical approval from the Ethics Committee of the Faculty of Behavioural and Movement Sciences, VU University Amsterdam, the Netherlands (ref. number VCWE-2019-139).

### 2.2. Assessment

*Actual Motor Competence*. AMC was measured with the Athletic Skills Track (AST). The AST is an age-specific track consisting of a series of 5–7 detached activities based on coordinative abilities (i.e., coupling, spatial orientation, and balance ability) to be completed as quickly as possible [57]. In this study, AST-1 (age group 4–6 years old) and AST-2 (age group 6–9 years old) were used with the difficulty of the tasks ascending from AST-1 to AST-2. AST-1 consisted of the following five fundamental movement skill tasks: (1) walking, (2) traveling jumps, (3) alligator crawl, (4) slaloming, and (5) clambering. AST-2 consisted of (1) walking, (2) traveling jumps, (3) hopscotch, (4) alligator crawl (backward), (5) running (backward), (6) pencil roll, and (7) clambering. The time to complete the track was registered to the nearest 0.1 second by a research assistant using a stopwatch. The test–retest reliability of the AST was proven to be high (ICC = 0.881 (95%), CI: 0.780–0.934) for AST-1 and AST-2 (ICC = 0.802 95%, CI: 0.717–0.858) in a sample of 4–12-year-old Dutch children [58]. The internal consistency was above the acceptable level of Cronbach’s α > 0.70 (AST-1: α = 0.764; AST-2, α = 0.700), and the validity of the AST was high when correlated with the Körperkoordinations Test für Kinder (KTK) (AST-1: r = −0.747, P = 0.01; AST-2: r = −0.646, P = 0.01) [58].

*Perceived Motor Competence*. To measure PMC, the 12-indicator Pictorial Scale of Perceived Movement Skill Competence for Young Children (PMSC) was used [59]. The children were individually assessed by the research assistant for six locomotor (LOC) skills and six object control (OC) skills using a pictorial instrument [59,60,61]. Children were shown a page with two cartoon illustrations of children undertaking each skill, one better than the other, and asked to identify which child is most like them (i.e., ‘this child is pretty good at throwing, this child is not that good at throwing, which child is most like you?’). Once selected, choices were further refined by asking ‘are you really good at …’ (scored as 4 points) or ‘pretty good at …’ (3 points) if the competent picture was selected; or ‘are you sort of good at …’ (2 points) or ‘not too good at …’ (1 point), if the ‘not so competent’ picture was selected. Test–retest reliability scores of the PMSC for 4- to 7-year-old children were excellent (ICCs for all 12 skills: 0.83, the six LOC items: 0.82 and the six OC items: 0.78). The internal consistency of the PMSC was above the acceptable level of Cronbach’s α > 0.60 for LOC as well as OC skills, and high correlation coefficients were reported between the pictorial scores and the Test of Gross Motor Development-2 (TGMD-2) (r = 0.82–0.90) [59,61].

*Physical activity enjoyment*. In order to assess children’s enjoyment of being physically active, a ‘smileyometer’ was used. On a 5-point Likert scale ranging from one ‘not nice at all’ to five ‘very nice’, children were individually asked by a research assistant to point out one smiley that best matched the specific question. The questions, accompanied by corresponding images, contain four different contexts of PA: (1) PE lessons, (2) a playground at school, (3) a playground in their own neighborhood, and (4) organized sports in a sports club. Additionally, one sedentary context was assessed (i.e., watching television/tablet). The smileyometer has been widely adopted and applied in research studies with children [62,63,64]. It is easy to complete and requires no reading or writing skills from young children. The reliability and age effect of the smileyometer have been positively evaluated in young children [65].

*Sports participation*. Next to PA enjoyment, children were asked if they were participating in a sports club on a regular basis. Participating in swimming lessons to obtain a swimming diploma was not included as a sports club activity. Sports participation was assessed by showing a picture of an organized sports setting. If children said they were participating, a follow-up question was asked if they could tell what sport they were doing. 

*Weight status*. At the start of the physical education (PE) lesson, body height and body weight of the children were measured individually by a research assistant to the nearest 0.1 cm using a stadiometer (SECA 217, Hamburg, Germany) and to the closest 0.1 kg using a digital scale (SECA 878dr, Hamburg, Germany). Children were measured without wearing shoes.

### 2.3. Procedure

Data collection took place in a regular 50-minute PE lesson during school hours. The lesson started with a general introduction by the PE teacher. Additional explanations and demonstrations were given by the research assistants at the specific test item. To minimize the emphasis on measuring, the children were instructed to play regular PE activities. The children were called in small groups to perform a specific test with the research assistant. The children wore regular sportswear and were barefoot during testing. The questionnaires to measure PMC, PA enjoyment, and sports participation started with a short introduction and an example. When the child understood, the research assistant read out the real questions to the child individually and filled in their answers. All research assistants participated in a two-hour training session to conduct the tests according to the protocol. Additionally, a supervisor was always present to ensure measurement quality and to organize the test setting.

### 2.4. Data Analysis

MC was expressed in age- and gender-specific motor quotient (MQ) categories. Based on the time to complete the AST, MQ-scores were generated with the following formula: MQ = (50th percentile AST/time AST) × 100 [58]. AST-time below the 25th percentile of AST norm values (corresponding with MQ categories *very low* and *low*) was categorized as ‘low’, an AST time between the 25th and the 75th percentile of the norm values (corresponding with MQ category *normal*) was classified as ‘normal’, and an AST time above the 75th percentile of the norm values (corresponding with MQ categories *high* and *very high*) was classified as ‘high’. Subsequently, patterns of MC development were defined based on the changes in MQ categories between T0 and T1 and between T1 and T2. The definition of an undesirable pattern is extensively described by Koolwijk and colleagues (2024) [14] and is based on a combination of the course of the absolute MQ scores and their categories over time.

The sum scores of PMC and PA enjoyment were categorized as ‘low’, ‘middle’, or ‘high’. PMC scores were calculated for LOC skills (*n* = 6) and OC (*n* = 6) skills separately. Sports participation was categorized as ‘yes’ (positively) or ‘no’ (negatively). BMI was calculated by dividing body weight (kg) by the square of the body height (m) while controlling for gender and age. Next, BMI was categorized into three meaningful categories, i.e., normal weight, overweight and obese, based on age- and gender-related cut off values [66]. High PMC, high PA enjoyment, normal weight status, and active sports club participation were included in the models as reference categories.

First, Spearman’s correlation coefficients were calculated for the correlations between PMC and PA enjoyment scores and to check for multicollinearity. To establish individual characteristics of AMC development, data were analyzed using linear mixed-effect (LME) models with continuous MQ scores on T0 and T2 as outcome variables. PMC, PA enjoyment, weight status, and sports participation were modeled as categorical exposure variables at both timepoints. Additionally, linear trends were explored for PMC, PA enjoyment, and weight status. Subsequently, the interaction of each determinant with time, reflected by the timepoint of data collection, was assessed by comparing the model fit of an LMM including this interaction term with the same model without this interaction term using ANOVA. The results of univariate analyses reflect the effect estimates adjusted for age and gender. As PMC was modeled using the sum of scores of 6 elements, these analyses were adjusted for the number of missing elements. When significant interaction was observed, effect estimates were reported for T0 and T2 separately. All models were adjusted for age and gender since MQ scores are age- and gender-standardized scores.

For significant determinants of AMC, children were grouped according to the change in these determinants between T0 and T2. For example, the categorical change in the determinant ‘weight status’ could be grouped as ‘developed overweight’ (increase), or ‘developed a normal weight’ (decrease) or remained stable (stable). For each of these groups, the proportion of children with AMC development previously classified as ‘undesirable’ was calculated. Descriptive data analysis was performed with the Statistical Package for the Social Sciences (SPSS version 27.0, 64-bit edition, SPSS Inc., Chicago, IL, USA). *p* values ≤ 0.05 were considered statistically significant. Regression modeling using the lme4- package and visualizations were obtained with R studio (version 4.2.2, R Foundation for Statistical Computing, Vienna, Austria).

## 3. Results

In this longitudinal design, our original study population contained 1131 children. After removing outliers based on MQ scores and excluding participants who had not undertaken all measurements at T0 and T2, our study sample consisted of 721 children (50.2% boys). In this table, sum scores and categorizations of AMC and potential determinants (i.e., PMC, PA enjoyment, organized sports participation, and weight status) are presented for timepoints T0 and T2 (Table 1). PMC and PA enjoyment scores were high at both timepoints, with slightly more variance in PMC scores at T2. The PMC sum scores of LOC skills were moderately correlated with PMC OC skills sum scores (Spearman’s rho = 0.54 at T0 and 0.52 at T2, *p* < 0.01) and weakly correlated with the sum scores of PA enjoyment (Spearman’s rho = 0.16 at T0 and 0.14 at T2, *p* < 0.01). Since there was no multicollinearity, all potential determinants were entered in the LME models.

### Significant Determinants of AMC

Two determinants were significantly associated with the MQ score on T0 and T2: weight status and organized sports participation (Figure 1). More specifically, overweight, obesity, and no organized sports participation were longitudinally associated with lower MQ scores. Interaction with time was observed only for weight status, and this association increased in strength over time, β = −1.52 (95% CI = −6.71 to 3.66) for overweight and β = −1.41 (95% CI = −8.81 to 5.98) for obesity at T0, and β = −7.57 (95% CI = −12.38 to −2.76) for overweight and β = −10.15 (95% −17.73 to −2.570) for obesity at T2. Multivariate analyses showed that associations of weight status and sports participation with MQ scores remained significant after adjustment for variation in PMC and PA enjoyment.

The proportion of children with an undesirable pattern of MC development was elevated not only when children developed overweight or obesity over time (22.7%) but also in children who had decreased in weight status over time (26.5%) (Table 2). With regard to sports participation, the proportion of children with undesirable MC development was also elevated among children who did not participate in organized sports during the full study period, as well as children who stopped participating in sports during the follow-up measurement (24% in both groups).

## 4. Discussion

The purpose of this study was to examine individual characteristics of children that might explain the undesirable pattern of MC development during the transition from EC to MCD in this group of children. Almost 19% of the Dutch children showed an undesirable pattern in MC development from EC to MCD [14]. This percentage falls within the range of cross-cultural differences in children’s MC development in Europe [67,68]. In southern Europe, approximately 40% of MCD-aged children exhibit worrying MC development against 10% in northern parts of Europe. Our study results reveal that children showing an undesirable pattern of MC development can be recognized by their weight status and sports participation at baseline and undesirable changes between EC and MCD. Excessive weight (overweight and obesity) is associated with lower MQ scores over time. This is in line with previous studies carried out with children in early childhood [29,31,69] and middle childhood [34,67], as well as during the transition from EC to MCD [23,28]. An unexpected finding was that among a small group of children who decreased in weight status between EC and MCD an undesirable pattern of MC development was seen. A clear explanation is hard to find but may have to do with the chosen cut-off values for BMI classification. It might also be that a higher weight status in the past may negatively affect MC status followed by a delayed development later on. Another explanation may be that these children suffer from an undiagnosed coordinative movement disorder. And finally, it could also be that the neurological motor system needs to adapt after a biomechanical change, just like in puberty, when motor coordination is temporarily disturbed when adolescents suddenly grow [70].

In our study, an association was found between sports participation and MC development. In the literature, findings are inconsistent. This inconsistency may be due to the overall goal of sports participation for EC-aged children. While some studies emphasize the value of sports participation for learning motor skills and increasing MC [68,71,72,73,74,75], other studies emphasize the benefits of sports membership on psychological and social outcomes [76]. Our results emphasize that lacking or quitting participating in organized sports can be of risk for MC development. However, children may also withdraw from sports because of their poor motor skills. Therefore, by asking why children stop participating in organized sports, more information can be gathered, which can help sports clubs professionalize and respond to the needs of children. 

In summary, our hypotheses that an undesirable course of MC development is related to higher weight status and lower sports participation can be confirmed. The results of this study contribute to Robinson’s revised conceptual model [6], in which children’s BMI negatively predicts MC, and sports participation is positively related to MC development over time. 

In addition to these main results, other findings regarding potential determinants of MC development are highlighted. With regard to PMC, no significant associations were found between LOC and OC skills scores and MC development over time. As expected, in early childhood, children often confuse the wish to be competent with reality due to a misunderstanding of the content, which leads to overestimation [77]. From MCD, children are better capable of assessing their competencies, as they are more able to compare their abilities with peers based on cognitive development [78]. Within the current study, as children grow older (i.e., timepoint T2), there is considerably more variation in overall reported PMC scores, which suggests more cognitive abilities to reflect their competencies with peers. However, significant associations between PMC/MC during the transition from EC to MCD were not yet found, which is in line with a recent study by Niemistö and colleagues (2023) [27].

Finally, the literature has shown that PA enjoyment positively influenced MC and prolonged PA behaviors throughout childhood [42,79]. However, in our data, no such relation was found. Therefore, this part of our hypothesis is rejected. A possible explanation for this finding could be the young age of our study population or the type of assessment used. Positive experiences and enjoyment of physical activities at a young age are important for prolonged sports participation. PE lessons at schools play an important role in providing these experiences. However, within our age category, most of the PE lessons are provided by generalists instead of PE professionals. Observations during PE lessons, conducted in a Dutch study by Adank and colleagues [80], showed that teaching practices given by PE professionals rather than by generalists are crucial for fostering PE enjoyment. Although this study was conducted with an older population (10–12-year-old children), the results could be applicable to younger children in early and middle childhood.

### 4.1. Limitations

Several important limitations should be considered when interpreting our findings. First, because of the COVID-19 pandemic, all children in the current study experienced a three-month lockdown with restrictions on PA at sports clubs in 2020. Refrainment from PE and sports participation could have impacted drop-out rates in organized sports participation and swimming lessons, as well as prevented children’s longitudinal MC development. Secondly, some methodological issues were present due to the assessments used in this study. Within this study, the 4-point (PMC) and 5-point (enjoyment) visual analog assessment tools were carefully chosen for alignment with our young study population. However, due to a lack of variance, group sizes were too small to perform meaningful regression analyses for the children with very low to low PMC or very low to lower PA enjoyment separately. Thirdly, the AST-1 and AST-2 are product-orientated rather than process-oriented MC assessment tools that focus on locomotor and balance skills. The Athletic Skills Track is a convenient, easy-to-administer, low-cost MC tool that can be used repeatedly in the PE setting [81,82]. This contributes to maintaining a high response rate in performing longitudinal research. However, the results might have been different for process-oriented MC values derived by other assessment tools (e.g., the TGMD-3, M-ABC, MOBAK test). In addition, object control skills were not determined. Fourth, with regard to research bias, some remarks need to be made. A significant sample of the original study population was excluded from our analyses because they did not complete all the measurements. Since the distribution of MQ scores in our current study population is similar to our previously conducted longitudinal study, there is no selection bias. 

### 4.2. Practical Implications and Future Perspectives

The outcomes of our study highlight the importance for sports and PE professionals to have a child-centered approach to support healthy MC development. Besides high-quality PE lessons and sports programs, children’s weight status should be monitored on a regular basis. In the Netherlands, initiatives are growing for schools in low-income neighborhoods to consult a public health advisor or a nutritionist to help create a healthy school policy and provide parental support for questions related to children’s health, physical activity, and nutrition. It is also the responsibility of PE teachers and other sports professionals to have a clear overview of the sports network (e.g., sports clubs, playgrounds, and other PA facilities in the neighborhood) of the school population. Eventually, helping young children to orientate on the wide range of sports activities meeting their needs will motivate children to join local sports, PA, and play initiatives.

Further research is needed to better understand long-term MC development from EC to MCD and to adolescence in a changing society where sedentary behavior is growing. Besides long-term research, differences in MC development may be due to several individual and environmental factors, such as the time allocated to PE during school, the physical home environment, and governmental policy regarding organized sports participation. In this study, PA levels have not been registered. It is, however, interesting to incorporate PA measurements via accelerometery to determine how levels of moderate to vigorous physical activity are associated with undesired MC development and weight status. Besides the level of PA, sedentary behavior should also be taken into account, since this is also related to MC development [83]. It would also be good to examine the relationship between social and physical environmental determinants and MC development and to pay more attention to gender differences regarding MC development. In general, studies have shown that boys are better at OC skills, whereas girls may be more competent in stability-demanding tasks and LOC skills [84]. Also, boys tend to report higher PMC compared with girls [85], with differences between countries worldwide [86]. It would be interesting to determine which determinants explain differences in MC development between boys and girls. Finally, researchers and practitioners should develop targeted interventions focusing on the high-risk population highlighted in this study.

## 5. Conclusions

This longitudinal study provides valuable insights into individual characteristics that are associated with an undesirable development of MC during the transition from EC to MCD. Excessive body weight and lack of sports participation between EC and MCD increase the risk of an undesirable MC development. Sport professionals play a crucial role in stimulating MC development from an early age and should especially support this subgroup of children. Further longitudinal research is needed to detect other determinants of undesirable MC development from EC into MCD.

## Figures and Tables

**Figure 1 ijerph-21-01460-f001:**
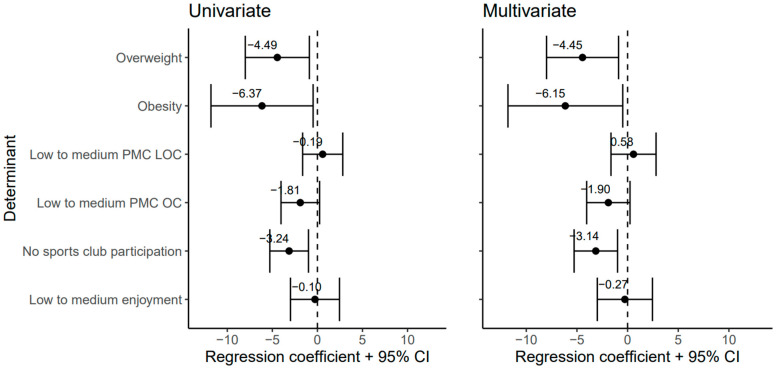
Univariate and multivariate associations of measured determinants: weight status (overweight and obesity), PMC (LOC and OC skills), sports participation, and enjoyment of PA, with the MQ scores. For each determinant, positive regression coefficients (β) reflect associations with higher MQ score and therefore a higher motor competence, whereas negative regression coefficients reflect associations with lower MQ scores. Regression coefficients (and confidence intervals) of the fixed effects represent differences in MQ scores between specified determinant and the reference category.

**Table 1 ijerph-21-01460-t001:** Characteristics of AMC and potential determinants at timepoints T0 and T2 (N = 721).

	Timepoint							
	T0				T2			
**AMC**								
MQ score (Mean (SD))	102.74 (19.5)				102.64 (19.8)			
Categorical (*n* = 721/%) *								
*Very low*	55		8%		54	7%		
*Low*	106		15%		103	14%		
*Normal*	382		53%		381	53%		
*High*	123		17%		119	17%		
*Very high*	55		8%		64	9%		
**Individual determinants**								
**PMC per set of skills**								
	LOC		OC		LOC		OC	
Sum score (Median (IQR))	21	(18–23)	20	(17–22)	20	(17–22)	19	(17–21)
Categorical (*n*/%)								
*Low (sum score 6–12)*	15	2%	27	4%	9	1%	29	4%
*Middle (sum score 13–18)*	171	24%	246	34%	258	36%	314	44%
*High (sum score 19–24)*	535	74%	448	62%	454	63%	378	52%
**Enjoyment of PA** Sum score (Median (IQR)) Categorical (*n*/%)	14 (12–15)					14 (13–15)		
*Low (sum score 3–6)*	2		<1%		0	0%		
*Middle (sum score 7–11)*	131		18%		76	11%		
*High (sum score 12–15)*	587		81%		644	89%		
*Missing*	1		<1%		1	<1%		
**Weight status**								
BMI in kg/m^2^ (Median (IQR))	15.44		(14.63–16.6)		15.61	(14.59–16.8)		
Categorical (*n*/%)								
*Normal weight*	630		87%		600	83%		
*Overweight*	60		8%		64	9%		
*Obesity*	27		4%		24	3%		
*Missing*	4		1%		33	6%		
**Organized sports participation (*n*/%)**								
*Yes*	253		35%		521	72%		
*No*	465		64%		199	28%		
*Missing*	3		<1%		1	<1%		

* Categorical distribution of MQ scores of original study population (*n* = 1131): 9.7% very low, 14.6% low, 52.0% normal, 15.1% high, and 8.6% very high. AMC: Actual Motor Competence, MQ: Motor Quotient, PMC: Perceived Motor Competence, LOC: Locomotor skills, OC: Object Control skills, PA: Physical Activity, BMI: Body Mass Index.

**Table 2 ijerph-21-01460-t002:** Categorical changes in determinants by pattern of MC development.

Determinant	Categorical Changes over Time (T0–T2)	Children with an Undesirable Pattern of MC Development (%)
Weight status	Increase	22.7
	Decrease	26.5
	Stable	18.7
Organized sports participation	Engaged in sports participation	17.8
	Dropped out sports participation	24.3
	Stable (member)	16.9
	Stable (non-member)	24.2

## Data Availability

The data are available on request from the corresponding author at the Data Station Life Sciences of the Data Archiving and Networked Services (Dans). The data are not publicly available due to ethical reasons.

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
