# Peer review of "Characteristics of Children with an Undesirable Motor Competence Development During the Transition from Early to Middle Childhood: Results of a 2-Year Longitudinal Study"

_ijerph, 2024, doi:10.3390/ijerph21111460_

Round 1

Reviewer 1 Report

Comments and Suggestions for Authors

1- It is suggested to mention the duration of the study in the title. For example, a 2-year longitudinal study

2- On what basis was the sample size selected?

3- The gender ratio should be mentioned in the abstract.

4- The number of keywords seems high. It is better to have fewer of them.

5- The introduction is well written and examines various studies and models and finally points out the existing research gap.

6- On what basis are schools divided based on socioeconomic status?

7- To measure actual motor competence, why other famous tools such as TGMD-3 and others were not used? Please mention this in the limitations section and suggest that future research use other methods.

8- How were the tests done? Were they the only subjects and examiners or in front of other children? More details can be mentioned in the method section.

9- It is suggested to specify the process of conducting the study in the form of a flowchart.

10- The results and discussion section are well written and the reader can easily understand it.

Author Response

We thank you for your time and effort in reviewing our manuscript. The feedback has been valuable in improving the content of the manuscript. My co-authors and I are pleased to submit our revised manuscript titled “Characteristics of children with an undesirable motor competence development during the transition from early to middle childhood: results of a longitudinal study." for reconsideration for publication.  

The changes are highlighted in the attached manuscript by using bold text, and our point-by-point responses are given below in italics. 

Comments (1): It is suggested to mention the duration of the study in the title. For example, a 2-year longitudinal study 

Response 1: Thank you for the suggestion, the authors agree, the study title has been adjusted (line 3). 

Comment (2): On what basis was the sample size selected? 

Response 2: This has been added in the methods section. Sentence: “The participating schools were all internship schools of the connected universities.” (lines 137-138). 

Comment (3): The gender ratio should be mentioned in the abstract. 

Response 3: Thank you for the suggestion, the gender ratio has been incorporated in the abstract. (line 21). 

Comment (4): The number of keywords seems high. It is better to have fewer of them. 

Response 4: We agree. We therefore have skipped the keyword longitudinal. (lines 32-33).   

Comment (5): The introduction is well written and examines various studies and models and finally points out the existing research gap. 

Response 5: Thank you for the compliment! 

Comment (6): On what basis are schools divided based on socioeconomic status? 

Response 6: Socioeconomic position scores are described in terms of the financial prosperity, education level and recent employment history of private households on January 1 of the reporting year. These data have been retrieved from the National governmental agency for statistics (ref. 56). Based on the zip codes of all participating schools (N = 36) socio economic position scores were derived and checked for their distribution (ranging from low: <-0.2 to high: >0.2). For clarification, this had been added to the sentence starting at line 132 “These data were gathered….history of households)[56]” 

Comment (7): To measure actual motor competence, why other famous tools such as TGMD-3 and others were not used? Please mention this in the limitations section and suggest that future research use other methods. 

Response 7: We specified the advantages of the AST assessment tool (lines 387-391) to legitimize the use of this assessment tool (supported by reference numbers 81,82). Next, we explicitly mention the value of process-orientated assessment tools such as the AST instead of the more common product-oriented tools like the TGMD-3, the Movement ABC and the MOBAK test (lines 391-392).  

Comment (8): How were the tests done? Were they the only subjects and examiners or in front of other children? More details can be mentioned in the method section. 

Response 8: More details have been added in the procedure section (lines 215-217). 

Comment (9): It is suggested to specify the process of conducting the study in the form of a flowchart. 

Response 9: Thank you for this suggestion. While writing this manuscript the authors have considered adding a flowchart (or descriptive table). However, we eventually chose not to add an extra illustration (flowchart/table) because it would lead to less attention to the main findings and message of our study.  

Comment (10): The results and discussion section are well written and the reader can easily understand it. 

Response 10: Again, thank you for reviewing our manuscript!

Reviewer 2 Report

Comments and Suggestions for Authors

This report is a well-written and provides adequate background literature with a good introduction, results, and discussion of findings.  Being a longitudinal study has merit. Motor competence is a timely topic. I do have the following notes:

1- Your title reflects your key finding " Characteristics of children with an undesirable MC." while your objective was " to examine which individual characteristics are associated with motor competence development during the transition from early to middle childhood.” I would consider (suggestion only) using your objective statement as a title.

2- Your range of ages is 4-7.5 years.. saying middle childhood 6-12 years), is a stretch. Maybe say early childhood or young children.

3-  I (and perhaps others) are not familiar with the Athletic Skills Track (AST) assessment for MC. Thus making it a bit difficult to compare with other studies.There was no supportive papers to review - just the the  normative and validation papers. 

4- Overall, this is a good descriptive account of the change in MC over a 2-year period. Data that add to the growing body of information. "Excessive body weight and lack of sports participation from early childhood are associated with an increased risk of an undesirable motor competence development over time." With that said, your team needs to make a better case as to what is new (contributing) information. That result is somewhat expected and not new.

  What about PA outside of sports? hrs per week.. Questionnaire.

Author Response

We thank you for your time and effort in reviewing our manuscript. The feedback has been very valuable in improving the content of the manuscript. My co-authors and I are pleased to submit our revised manuscript titled “Characteristics of children with an undesirable motor competence development during the transition from early to middle childhood: results of a longitudinal study." for reconsideration for publication.  

The changes are highlighted in the attached manuscript by using bold text, and our point-by-point responses are given below in italics. 

This report is well-written and provides adequate background literature with a good introduction, results, and discussion of findings.  Being a longitudinal study has merit. Motor competence is a timely topic. I do have the following notes: 

Comment (1): Your title reflects your key finding " Characteristics of children with an undesirable MC." while your objective was " to examine which individual characteristics are associated with motor competence development during the transition from early to middle childhood.” I would consider (suggestion only) using your objective statement as a title. 

Response 1: Thank you for this suggestion. We added the duration of the study to the title as suggested by another reviewer to be more accurate and changed the objective of the study (lines 17-18 and 123-124). 

Comment (2): Your range of ages is 4-7.5 years.. saying middle childhood 6-12 years), is a stretch. Maybe say early childhood or young children. 

Response 2: Age ranges of children in early-, middle- and/or late childhood differ between studies. Based on literature, we defined early childhood children from the age range of 2-5 years old and middle childhood children from 6-9 years old. Therefore, our study population covers the transition phase from early- to middle childhood.  

Comment (3): I (and perhaps others) are not familiar with the Athletic Skills Track (AST) assessment for MC. Thus, making it a bit difficult to compare with other studies. There was no supportive papers to review - just the normative and validation papers.  

Response 3: The Athletic Skills Track is a well-known test for identifying levels of MC in the Netherlands, especially in the PE setting at primary schools. It has also been used in Australia (ref. 82) (Klingberg et al., 2019). The authors are aware of its rather unknown international status. We therefore specified the AST test by adding the advantages of this assessment tool (lines 387-391). We have added that the results can differ for other, more familiar, assessment tools like the TGMD-3, Movement-ABC, MOBAK.  (lines 391-392).   

Comment (4): Overall, this is a good descriptive account of the change in MC over a 2-year period. Data that adds to the growing body of information. "Excessive body weight and lack of sports participation from early childhood are associated with an increased risk of an undesirable motor competence development over time." With that said, your team needs to make a better case as to what is new (contributing) information. That result is somewhat expected and not new. 

What about PA outside of sports? hrs per week.. Questionnaire.  

Response 4:  

We agree that the results are in line with expectations. However, based on Robinson’s revised conceptual model (2015) more (longitudinal) research needed to be conducted especially from early childhood aged children. In addition, the number of studies implementing various determinants of MC development together are scarce. So, we think that this study adds valuable information to MC development in general.  

For getting more insight in the exact amount of PA by early childhood children, authors would like to refer to a (recently) published article:   

  • Remmers, T., Koolwijk, P., Fassaert, I., Nolles, J., de Groot, W., Vos, S. B., ... & Van Kann, D. H. H. (2024). Investigating young children’s physical activity through time and place. International journal of health geographics, 23(1), 12.  

In this study we used GPS/accelerometers to objectively monitor PA of early childhood children in a subsample. We do not have longitudinal data on the PA level of the sample described in the current manuscript, but we agree with the reviewer that this is interesting to examine further in future studies.